# SAGE: Semantic-Aware Global Explanations for Named Entity Recognition

## Abstract

In the last decades, deep learning approaches achieved impressive results in many research fields, such as Computer Vision and Natural Language Processing (NLP). NLP in particular has greatly benefit from unsupervised methods that allow to learn distributed representation of language. On the race for better performances Language Models have reached hundred of billions parameters nowadays. Despite the remarkable results, deep models are still far from being fully exploited in real world applications. Indeed, these approaches are black-boxes, i.e. they are not interpretable by design nor explainable, which is often crucial to make decisions in business. Several task-agnostic methods have been proposed in literature to explain models' decisions. Most techniques rely on the "local" assumption, i.e. explanations are made example-wise. In this paper instead, we present a post-hoc method to produce highly interpretable global rules to explain NLP classifiers. Rules are extracted with a data mining approach on a semantically enriched input representation, instead of using words/wordpieces solely. Semantic information yields more abstract and general rules that are both more explanatory and less complex, while being also better at reflecting the model behaviour. In the experiments we focus on Named Entity Recognition, an NLP task where explainability is under-investigated. We explain the predictions of BERT NER classifiers trained on two popular benchmarks, CoNLL03 and Ontonotes, and compare our model against LIME (Ribeiro et al., 2016) and Decision Trees.

*Keywords: Explainable AI, XAI, Explainable ML, Named Entity Recognition.*

## 1 Introduction

In recent years, Artificial Intelligence (AI) algorithms, especially deep learning models, are emerging in many applications, reporting state-of-the-art performances in many fields. In NLP, for example, the use of Large Language Models (LLM) based on huge deep neural networks achieved impressive results in many linguistic tasks. However, despite the remarkable results, deep approaches are still far from being fully exploited in real world applications. One major issue is the lack of interpretability and control of the models' predictions. This is often an important requirement for many industrial applications, especially in domains like medicine, defense, finance and law, where it is crucial to understand the decisions and build trust in the algorithms.

The increasing need to address the problem of interpretability and improve model transparency made the "Explainable Artificial Intelligence" a very popular research area in the Computer Science world. Explainable AI (XAI) or Interpretable AI or Explainable Machine Learning (XML) (Guidotti et al., 2021) is a broad area of research that studies and proposes AI approaches where humans can understand the causes underlying the decisions and predictions made by the machine (Vilone & Longo, 2021b). The AI algorithms can be usually grouped into two families (Vilone & Longo, 2021a): (a) white-box models, which include algorithms whose interpretation is given by design, and (b) black-box approaches where, on the other hand, the decision making process is "opaque" and hard to understand. White-box models such as linear regression, probabilistic classifiers or decision trees are significantly easier to explain and interpret, but, often, provide a low predictive capacity and they are not always capable of modeling the inherent complexity of the task. In black-box models, on the other hand, very little knowledge is available on how the input variables influence the final decision. The relationship between input and output is often the result of a complex composition of

mathematical functions and is not directly interpretable. Although classic ML approaches are still widespread, almost all the modern complex AI techniques, such as deep neural networks, are naturally opaque (Lipton, 2018). Thus many new methods aimed to make new models more explainable and interpretable have been proposed or are under investigation.

Most explainability techniques rely on the "local" assumption where the descriptions are provided for each example and few approaches exist aiming at provide a interpretable description of the model as whole (glass-box). In this paper, we present SAGE (Semantic-Aware Global Explanations), a method to produce highly interpretable global rules to explain NLP classifiers. Rules are extracted using a data mining algorithm which exploits a semantically enriched input representation. Semantic information yields more abstract and general rules that are both more explanatory and less complex, while being also better at reflecting the model behaviour. In the experiments, we focus on Named Entity Recognition, an NLP task where explainability is under-investigated. In particular, we aimed to explain the predictions of a BERT-based ((Devlin et al., 2018)) NER classifier trained and tested on two popular benchmarks, CoNLL03 (Tjong Kim Sang & De Meulder, 2003) and Ontonotes (Pradhan et al., 2013). We compare the proposed model against LIME (Ribeiro et al., 2016), which currently is one of the most popular local explanation algorithm in NLP and Decision Tree classifiers that are classic explainable by design, global explanation models.

For the assessment, we exploited two commonly used metrics: fidelity and complexity. The results show that the proposed approach infers a set of rules that reproduce the behavior of the model more accurately than both LIME and Decision Trees.

The paper is organized as follow. In Section 2, we summarize related works, while in Section 3 the proposed algorithm is described in detail. Experiments are reported in Section 4, and finally conclusions and future works are drawn in Section 5.

## 2 RELATED WORK

As deep learning models have become more complex, many methods have been proposed to interpret and explain the predictions of a model. Two main groups of XAI techniques exist: (1) local approaches, which aim to provide an interpretable explanation for each single prediction; (2) global approaches, which try to build a "white-box" version of the black-box model (thus interpretable by design).

**Local Explanations.** Local algorithms focus on finding an interpretable explanation of the prediction returned by the ML model for a given input example. Several approaches focus on interpreting the internal components of a black-box model with intent of shading some lights on its decision making process. In (Csiszár et al., 2020), authors propose the use of fuzzy logic to "explain" each Artificial Neural Network unit. Similar techniques try to identify which features in a particular input vector contribute the most to a neural network's decision. Layer-wise relevance propagation (LRP), for example, exploits a back-propagation similar algorithm to build a heat-map over the input features (Binder et al., 2016),(Montavon et al., 2019). This method proved to be very effective in Computer Vision, highlighting which pixels of an image contributed most to the final classification. Other methods such as (Lundberg & Lee, 2017) and (Simonyan et al., 2013) follow the same principles. Such techniques, however, are strongly related to the ML models' family used and they can be employed only with neural networks. Furthermore they usually do not produce high-level explanations.

Some model-agnostic approaches were proposed in (Ribeiro et al., 2016; 2018; Strumbelj & Kononenko, 2010). In particular, in (Ribeiro et al., 2016), LIME (Local Interpretable Model-agnostic Explanations) is presented. LIME trains a white-box classifier, that learns the black-box output distribution on a neighborhood of a given example. Neighbors are obtained perturbing the input features (usually randomly). The explanation can be then constructed by selecting the input features that mostly affect the model prediction. Although these models have been widely applied in explainability problems, including NLP, some limitations occur when dealing with Large Language Models (LLMs) and Named Entity Recognition. LLMs tokenize text in subwords, like wordpieces (Schuster & Nakajima, 2012) in BERT, which requires particular attention in designing the text perturbation step. Masking and perturbation of the input introduce another issue for LLMs, that are contextual. Indeed random masking of tokens can produce artificial and inconsistent contexts, which

may invalidate the identification procedure of significant features. Finally, performing hundreds of input perturbation per sample, can be an expensive process, especially in problems like Named Entity Recognition (NER), where the number of examples corresponds to the number of tokens in a dataset.

Most of the interpretable or explainable models for NLP is on text classification problems (Chen & Ji, 2020; Chen et al., 2021). For instance, (Alvarez-Melis & Jaakkola, 2017) proposed a method for interpreting sequence-to-sequence models in Machine Translation, whereas (Tuan et al., 2021) focused on local explanations in dialog generation. To the best of our knowledge, we are the first investigating XAI techniques in NER.

**Global Explanations.** While the explanation of a single prediction provides the user with a rationale for the classifier's behavior, it is not sufficient to convey confidence in the model as a whole. Thus, in contrast to local approaches, global explanation algorithms aim to find an interpretable description of the model in the whole input space.

White-box models, such as Decision Trees (Breiman et al., 2017) and Bayesian Rule Lists (BRLs) (Letham et al., 2015), are explainable by design models that are used to provide global explanations. Alternatively, one could extrapolate global rules from local explanations as in (Ribeiro et al., 2016), where authors proposed an extension of LIME for making global explanations. The resulting SP-LIME algorithm is based on the construction of an explanation matrix which collects the local importance of the interpretable components for each instance in a dataset $X$. For each component, a global importance in the explanation space is evaluated and the "Submodular pick (SP)" algorithm selects a subset of instances with the highest marginal coverage gain.

Our proposed method (SAGE) falls in this family. Unlike SP-LIME, it is natively global by design and aims to identify the set of semantically-enriched associative rules that best cover the behavior of the model on a reference set (fidelity). Our algorithm shares some principles behind BRLs, however, the mining process is specifically designed and optimized to scale in NLP problems like NER. Furthermore, differently from BRLs, SAGE is not used as an interpretable model per-se, but it is rather an algorithm to explain globally black-box classifier predictions. To the best of our knowledge, SAGE is the first method involving semantic knowledge in the generation of the explanations.

## 3 SEMANTIC-AWARE GLOBAL EXPLANATIONS

In this Section, we present SAGE (Semantic-Aware Global Explanations) algorithm, a method to discover globally significant explanation rules. To find a global and interpretable description of a black-box classifier, the proposed method follows a data-mining approach. In particular, given a black-box model $\Theta_{BB}$, the general idea is to apply an Association Rules Discovery algorithm to the learning set coupled with the model's predictions to find the most representative set of explanation rules for the original model. Although it is task agnostic, we present SAGE by exploiting the NER task. Since we are facing a NLP task, the algorithm employs semantic features to improve the generalization capabilities of the returned rules. A sketch of SAGE method is depicted in Figure 1. In this section, firstly we discuss how to formulate NER in the scope of data-mining task. Next, we

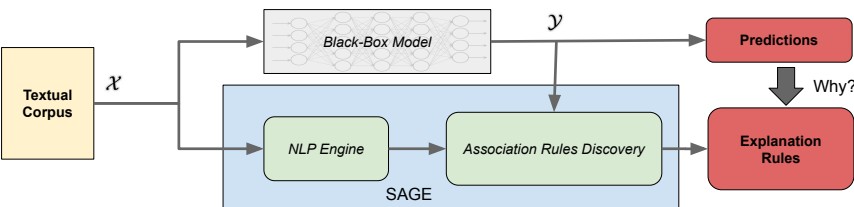

Fig. 1: Sketch of Semantic-Aware Global Explanations (SAGE) method.

delineate the data mining algorithm to generate candidate rules and the strategies for pruning and selecting a restricted number of explanations.

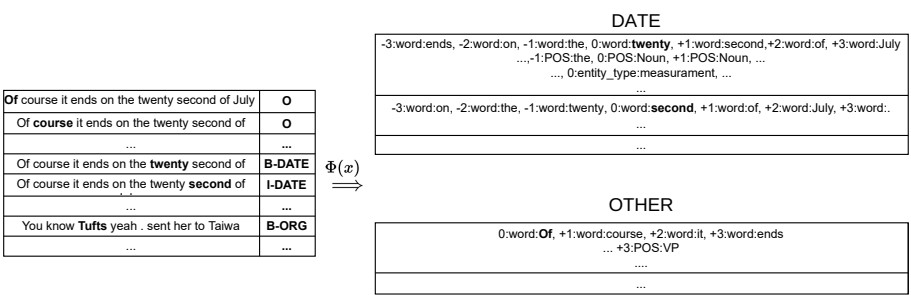

Fig. 2: Transactions creation. The function $\Phi$ produce a transaction for each token $x$ in the corpus. In particular, the example shows the transactions created for the three tokens **twenty**, **second** and **Of**. In the example, $\Phi$ considers a $(-3, +3)$ context window and includes semantic/syntactic information for each token in the window (e.g. the POS tags).

## 3.1 NER AS A DATA MINING PROBLEM

Named Entity Recognition (NER) is typically addressed as token classification task. Given a corpus of sentences/documents $\mathcal{X}$ and its corresponding annotations $\mathcal{Y}$, the goal is to predict a sequence of labels $\boldsymbol{y} \in \mathcal{Y}$, $\boldsymbol{y} := (y_1, \ldots, y_n)$, from a sequence of tokens $\boldsymbol{x} \in \mathcal{X}$, $\boldsymbol{x} := (x_1, \ldots, x_n)$.

In data mining, data is organized in a collection of transactions. A transaction $t_i$ consists in a set of $n_i$ items drawn from a specific basket $B$: $t_i = \{e_{i,1}, \ldots, e_{i,n_i}\}$, where $e_{i,j} \in B$ represents the $j^{th}$ element of the $i^{th}$ transaction. A transaction is often also referred to as "itemset". Given a large set of transactions $T = \{t_1, \ldots, t_m\}$, approaches as apriori (Agrawal et al., 1994) aim to discover frequent co-occurrence patterns (frequent itemset) in form of implications rules, e.g. $\{e_r, e_s, e_t\} \implies e_v$. A such rule states that when the premise is satisfied (i.e. the elements in the left part occur), the consequence should be considered (i.e. also the elements in the right part should occur).

Tackling NER with data mining requires to arrange textual data into transactions. We define the basket as the vocabulary containing the tokens from both sets $\mathcal{X}$ and $\mathcal{Y}$. Moreover, suppose each token $x_i$ is represented with a set of features obtained by a function $\Phi : \boldsymbol{x}, i \to \boldsymbol{t}$, then each feature can be seen as an item of the token transaction. Beside its representation, also the label $y_i$ is included in the basket. Consequently, we obtain a corpus of transactions $\mathcal{T}$ where each row is associated to its target label $(\Phi(\boldsymbol{x}, i), y_i)$. For the sake of simplicity, from now on we define $\Phi(x_i) := \Phi(\boldsymbol{x}, i)$. In its simplest form, function $\Phi$ is the identity, returning the token $x_i$ itself, or $x_i$ and a window of surrounding tokens. In general, it can be any function that returns a sequence of symbols. Since our goal is to obtain rules to predict NER classes, we are interested only in implication patterns where the conclusion is the target label $y_i$, i.e. $(\ldots) \implies y_i$ kind. This allows us to divide the problem into independent sub-problems, one per entity class in the corpus, on a reduced set of transactions $\mathcal{T}_{y_i}$. Since each class is treated independently, the generation of frequent patterns is dramatically simplified, which is a crucial aspect for the scalability of apriori-like methods to NER datasets.

Explaining a NER model is an analogous problem, the only difference is that labels $\mathcal{Y}$ are the output predicted by a classifier, instead of being provided by an oracle.

## 3.2 SEMANTIC FEATURES

Instead of simply using plain words or wordpieces as transaction items, we enrich them with semantic and syntactic positional information, extracted by $\Phi(x_i)$ that is a NLU system as shown in Figure 2. We emphasize the importance of having more general rules because they are more human-readable and reduce the overall number of explanations required to explain a model. We believe that they are essential to make explanations scalable in language, where data is sparse, especially in NER, and information is also positional. For the NLP analysis we employed a proprietary NLP engine[1] and we extract POS tagging information, entity types, concepts (e.g. synsets) and classical NER features such as *is_digit, is_upper_case, is_title* that are often important named entities indicators.

---

[1]The expert.ai NLP Platform - `https://www.expert.ai`

## 3.3 SAGE

SAGE is based on the FP-growth (Han et al., 2000) algorithm, an efficient variant of apriori (Agrawal et al., 1994). Due to the combinatorial nature of the problem and the relatively large amount of items in the transactions, the algorithm considers several thresholds: (1) the minimun support threshold ($mingensup$) which is commonly used in all data-mining algorithms to limit the outdegree of the candidate generation; (2) a minimum confidence threshold ($mingenconf$) which, following (Wang et al., 2002), we introduced to retain the candidates generation to the most promising frequent patterns only; (3) the tolerance threshold ($\tau$) which we introduced to stop adding items to a rule when confidence improvement is below such value; (4) a maximum rule length threshold ($maxrulelen$) which limits the length of generated rules to a maximum value; (5) a max number of rules to generate ($maxnrules$), which sets a maximum number of rules that can be generated. The algorithm is reported in Algorithm 1 and consists of three main procedures, each designed to face a specific step:

---

**Algorithm 1** Semantic-Aware Global Explanations

---

    **function** SAGE($\mathcal{X}, \mathcal{Y}$)
        **for** $y_i \in \mathcal{Y}$ **do**
            Given $\tau, mingensup_i, mingenconf_i, maxrulelen_i, maxnrules_i$
            $\mathcal{T}_{y_i}, \mathcal{T}_{\neg y_i} \leftarrow$ GET-TRANSACTIONS($\mathcal{X}, y_i$)
            $R \leftarrow$ CANDIDATES-GEN($\mathcal{T}_{y_i}, \mathcal{T}_{\neg y_i}, \tau, mingensup_i, mingenconf_i, maxrulelen_i$)
            $R \leftarrow$ PRUNE($\mathcal{T}_{y_i}, \mathcal{T}_{\neg y_i}, R$)
            $E_{y_i} \leftarrow$ SELECTION($R, niters, maxnrules_i$)
        **end for**
    **end function**

---

### 3.3.1 CANDIDATES GENERATIONS

The core of data-mining algorithms is the candidates generations step. At each iteration, the function CANDIDATES-GEN($\cdot$) produces a new set of frequent itemsets for the next step, starting from current set of transactions and using the transaction dataset as statistical reference. In particular, we exploited FP-growth (Han et al., 2000), an efficient variant of apriori (Agrawal et al., 1994).

### 3.3.2 RULES PRUNING

Rules mining can produce many patterns that are often spurious and/or highly overlapping. The over-generation of rules harms both the quality and the interpretability of the explanations. As a matter of fact, simplicity is crucial to deliver clear explanations that humans can easily understand. Thus, we aim to generate a minimal rule set with limited overlaps and high faithfulness. We proceed in two steps: pruning and selection.

In the pruning procedure , we start from the observation that most of the rules co-occur in the same subset of baskets, making them pseudo duplicate rules. Hence, among them we prune less generic rules, i.e. the ones having lower support. In case of rules that triggers in the exact same set of transactions, we favor more abstract rules, i.e. the ones having less conditions and/or more high-level concepts as items. Pruning step typically drops the number of relevant patterns by an order of magnitude[2]. Pseudo code is presented in Algorithm 2.

### 3.3.3 RULES SELECTION

After pruning, a large number of candidates still remains. Most of them have high overlap in the corpus. In the second step we aim to keep the optimal subset $S^* \in S$ of rules that provides the best performances. The problem is combinatorial - there are $2^{|S|}$ subsets - thus, we perform a greedy strategy. For a specified number of iterations $n_{iters}$, we filter candidates based on their support and confidence and then sub-select at most $maxnrules$ such that they maximize a performance metric. Support and confidence thresholds are sampled with bayesian optimization. Rules sub-selection is performed greedily: the procedure starts from an empty state (no rules solution) and adds one rule

---

[2]See Appendix A for further details.

---

**Algorithm 2** Rules pruning

---

**function** PRUNE($\mathcal{T}_{y_i}, \mathcal{T}_{\neg y_i}, R$)
    $R_{pruned} \leftarrow \emptyset$
    $R \leftarrow$ SORT-BY-SUPPORT-AND-ABSTRACTION($R, descending$)
    **for** $r \in R$ **do**
        $\mathcal{T}_r \leftarrow$ SUPPORT-TRANSACTIONS($r, \mathcal{T}_{y_i}, \mathcal{T}_{\neg y_i}$)
        **if** $\nexists p \in R_{pruned} | \mathcal{T}_r \subseteq$ SUPPORT-TRANSACTIONS($p, \mathcal{T}_{y_i}, \mathcal{T}_{\neg y_i}$) **then**
            $R_{pruned} \leftarrow R_{pruned} \cup r$
        **end if**
    **end for**
    **return** $R_{pruned}$
**end function**

---

at a time, storing the configuration that provide the higher improvement to the state. Despite the approach is heuristic, it improves the quality of the solution, both in terms of fidelity and complexity reduction as depicted in the section 4. Selection procedure is described in Algorithm 3.

---

**Algorithm 3** Explanations selection

---

**function** SELECTION($R, niters, maxnrules$)
    $S^* \leftarrow \emptyset$
    **for** $i = 1, \cdots, niters$ **do**
        $supp, conf \leftarrow$ BAYESIAN-HYPERPARAM-OPTIMIZATION($supprange, confrange$)
        $R \leftarrow$ FILTER($R, supp, conf$)
        $S \leftarrow \emptyset$
        **while** $|S| < maxnrules$ and $|S| < |R|$ **do**
            $r \leftarrow \underset{r \in R \setminus S}{\arg\max}$ COMPUTE-FIDELITY($S \cup r$)
            **if** COMPUTE-FIDELITY($S \cup r$) $\leq$ COMPUTE-FIDELITY($S$) **then**
                **break**
            **end if**
            $S \leftarrow S \cup r$
        **end while**
        **if** COMPUTE-FIDELITY($S$) $>$ COMPUTE-FIDELITY($S^*$) **then**
            $S^* \leftarrow S$
        **end if**
    **end for**
    **return** $S^*$
**end function**

---

## 4 EXPERIMENTS

We evaluate SAGE in two different NER benchmarks, aiming to measure (1) the quality of the explanations, (2) how semantic information is vital to produce generalized rules that are both more effective and easier to understand.

### 4.1 METRICS

For evaluating the explanations, we consider two popular criteria: fidelity and complexity.

**Fidelity.** Fidelity measures the ability of the explanation algorithm to mimic the predictions of the black-box classifier (e.g. BERT). To measure such index, we considered the F1 score (both micro and macro). In this setting, the optimal value $F1 = 100\%$ means that the explainable model perfectly reflects the behavior of the black-box model.

**Complexity.** Complexity on the other hand, measures the simplicity of resulting explanations. We define it as the number of rules per class extracted by the data-mining algorithm.

## 4.2 DATASETS

**Ontonotes.** Ontonotes[3] is a dataset containing named entity annotations for multiple languages. There are multiple versions, in the experiments we used English v12 (Pradhan et al., 2013). The corpus is composed of texts from various genres. Overall, there are 18 annotated classes.

**CoNLL03.** CoNLL03[4] (Tjong Kim Sang & De Meulder, 2003) is a standard Named Entity Recognition (NER) benchmark. The dataset is a collection of news articles from the Reuters corpus. Data is annotated with four entity tags: PERSON, LOCATION, ORGANIZATION and MISC.

## 4.3 EXPERIMENTAL SETUP

**Black-box classifiers.** We train a BERT (Devlin et al., 2018) classifier on each benchmark (CoNLL, Ontonotes) and we explain its predictions on the test set. Predictions are made at word level, i.e. we ignore the output of word sub-tokens as commonly done in NER problems with models based on word-pieces like transformers. Overall, we obtain $4170$ and $14227$ predicted entities in CoNLL and Ontonotes, respectively.

**LIME for NER.** LIME is widely used for explaining models in classification problems. However, there are some caveats when applying it in NER. Indeed, predictions are at token-level, while usually the approach is applied for document-level classification problems. Thus, explanations must be positional. Furthermore, current Language Models adopt subword-level tokenizers, while we expect to have word-level explanations. Hence, wordpieces of the same words are perturbed together. For each example, we retrieve the top-$k$ words to build an explanation. Global explanations are then constructed applying submodular-pick (SP-LIME-$k$). In the experiments, we consider five different SP-LIME models differentiated by the choice of $k$ that varies from 1 up to 5.

**Decision Tree.** Decision Trees (DTs) are considered to be white box classifiers, therefore they are widely used for explainability. We compare SAGE with DTs fitted with (DT) and without (DT-words) the semantic features. We fit trees with maximum depth of 100.

**SAGE.** Transactions are constructed as described in Section 3. Function $\Phi$ produces for each token $x_i$ a set of positional features in a $(-3, +3)$ context window, as presented in Section 3.2 and depicted in Figure 2. In the results we also report performances of a variant, called SAGE-words, that exploit a simpler $\Phi$ function that yields only words in the $(-3, +3)$ scope, i.e. without any semantic information. Hyper-parameters of Algorithm 1 could be set differently for each entity. In the experiments, however, most of the hyper-parameters are kept the same for all classes and on both datasets. In particular, we set $\tau = 0.05$, $maxrulelen = 5$, $niters = 100$ and $maxnrules = 100$ everywhere. Only $mingensup$ and $mingenconf$ change depending on the number of occurrences of the class. The minumum confidence to further explore a rule is set dinamically to $mingenconf_i = \frac{|\mathcal{T}_i|}{|\mathcal{T}|}$, whereas the minimum support is set to 10 for all entities in CoNLL, and varies between $\{2, 5, 15\}$ in Ontonotes depending on the cardinality of $\mathcal{T}_i$ and such that large classes have higher $mingensup$ and, vice versa, underrepresented entities have smaller $mingensup$.

## 4.4 RESULTS

**Quality of explanations.** We evaluate the quality of SAGE explanations in terms of fidelity and complexity as defined in Section 4.1. In particular, we measure how the increase of complexity of the explanations improves the fidelity to the original model, which may be useful to determine a satisfactory trade-off between the two indicators. Results are reported in Figure 3.

SAGE significantly outperforms SP-LIME global explanations. Even a single rule per-class outreaches any SP-LIME setup. Figure 4 shows the F1 scores for each single entity class in Ontonotes. We can see, that SAGE always outperforms SP-LIME explanations, with the exception of LANGUAGE and PRODUCT, two underrepresented classes where SAGE performs poorly because it cannot find rules that are statistically significant enough.

---

[3] https://huggingface.co/datasets/conll2012_ontonotesv5/
[4] https://huggingface.co/datasets/conll2003.

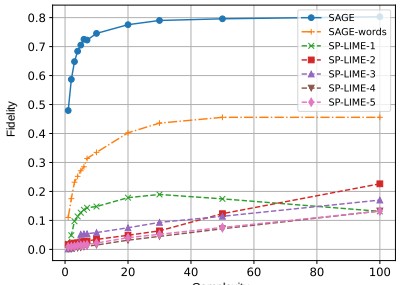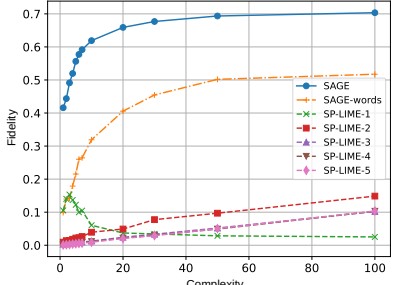

Fig. 3: Explanation performances of our method (with and without semantics) and different models built with SP-LIME. We compare all methods in terms of $\mu$-fidelity score at the varying of explanation complexity (number of explanations per class) on CoNLL03 (left) and Ontonotes (right).

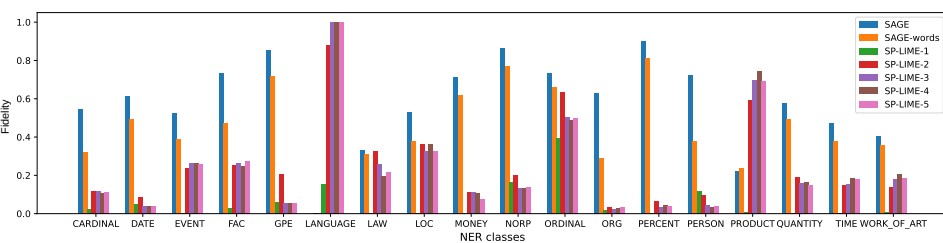

Fig. 4: Class-wise $\mu$-fidelity scores for each approach in Ontonotes dataset.

There are several reasons responsible for causing such a gap between SAGE and SP-LIME. First of all, LIME is a local algorithm designed to provide local explanations of predicted examples, and SP-LIME combines them together to provide an overall view of them. However, each explanation is not designed to hold outside the specific example, whereas our approach aim to retrieve statistically relevant rules that trigger a class in the data. Furthermore, token classification tasks, such as NER, introduce additional complexity because explanation must be positional, and because most of the tokens do not belong to an entity. Lastly, as we will see shortly, semantics is vital to boost performances of our approach.

One could notice however, that SP-LIME-$k$, $k > 1$ have a linear growth as the number of explanations increases, and it may eventually surpass SAGE performances, $complexity \rightarrow \infty$. As a matter of fact, this will happen, because by construction SP-LIME will retrieve one explanation per example, falling back to providing local explanations. This situation occurs in Ontonotes for class LANGUAGE. Looking at Figure 4, we can see that SP-LIME methods are performing extremely well on such entity, because the number of annotated tokens is only 22, hence SP-LIME extracts one rule per example. However, the number of explanations to reach and surpass SAGE is extremely high, breaking even far from a point where the explanation is easy to understand.

SAGE also outperforms Decision Trees, as outlined in Table 1, also providing rules that are naturally more human-friendly than tree decision paths that may involve dozens of conditions.

**The impact of semantics.** Semantic information plays a crucial role in providing better rules. We analyse this aspect comparing SAGE with SAGE-words, a variant where a token $x_i$ is represented only by itself and text in its context, without any external knowledge. Results are outlined in Table 1 and Figure 3.

In all the scenarios, semantics provides a significant gain in terms of fidelity. One explanation per class in SAGE outperforms 100 rules of SAGE-words. Indeed, abstraction is a mean to group less general concepts into a single one. It allows to express rules with higher coverage, and it is particularly important for language, that is characterized by the presence of many discrete symbols (words) combined together. A clear example of how semantics yields more general rules can be

| Explainer | CoNLL03 | | Ontonotes | |
|---|---|---|---|---|
| | μ-fidelity | m-fidelity | μ-fidelity | m-fidelity |
| DT-words | 34.8 | 34.2 | 7.4 | 17.3 |
| DT | 76.0 | 74.5 | 32.1 | 32.6 |
| SAGE-words | 45.6 | 40.0 | 51.5 | 44.7 |
| SAGE | **80.3** | **78.5** | **70.3** | **57.4** |

Tab. 1: Evaluation of explanations in CoNLL03 and Ontonotes datasets for our approach and Decision Trees (DT), both with and without semantic features.

seen in Figure 5. SAGE rules exploit broader concepts that are valid for much more examples.
Furthermore, the effectiveness of semantics is independent from the explainer algorithm. Even decision trees benefit from it (see Table 1).

| | CARDINAL | DATE | EVENT |
|---|---|---|---|
| **SAGE** | **pos**:ADJ ∧ word:one | **synset**:definite time | 1:**ent**:EVN ∧ **pos**:ART ∧ 3:**istitle** |
| **SAGE-words** | word:two ∧ -1:word:the | word:year | 1:word:Year ∧ word:New |
| **SP-LIME-1** | word:two | 1:word:year | word:Katrina |
| | **FAC** | **GPE** | **LANGUAGE** |
| **SAGE** | **istitle** ∧ 1:**synset**:road | **ent**:GEO | - |
| **SAGE-words** | 1:word:Road | word:China | - |
| **SP-LIME-1** | 3:word:Road | word:China | word:English |
| | **LAW** | **LOC** | **MONEY** |
| **SAGE** | 1:**synset**:bill of rights | **ent**:GEA ∧ **synset**:geographic area | **ent**:MON |
| **SAGE-words** | 1:word:Amendment | -1:word:Middle | -1:word:$ |
| **SP-LIME-1** | word:Gramm | 1:word:Middle | -1:word:$ |
| | **NORP** | **ORDINAL** | **ORG** |
| **SAGE** | **synset**:Asian | word:first | **ent**:COM |
| **SAGE-words** | word:Chinese | word:first | 1:word:Inc |
| **SP-LIME-1** | word:American | word:first | word:CNN |
| | **PERCENT** | **PERSON** | **PRODUCT** |
| **SAGE** | **ent**:PCT | **pos**:NPR.NPH | word:cole |
| **SAGE-words** | -1:word:. ∧ 1:word:% | -2:word:Mr | word:Cole |
| **SP-LIME-1** | 3:word:% | word:Bush | word:Cole |
| | **QUANTITY** | **TIME** | **WORK OF ART** |
| **SAGE** | **ent**:MEA ∧ -3:**pos**:NOU | **ent**:HOU | **istitle** ∧ **ent**:MEA |
| **SAGE-words** | 1:word:degrees | word:hour | 1:word:Minutes |
| **SP-LIME-1** | 1:word:degrees | 1:word:morning | word:Sixty |

Fig. 5: Explanation rules discovered by SAGE, SAGE-words and SP-LIME-1 in Ontonotes with complexity 1. Semantic features are highlighted in bold.

## 5 CONCLUSIONS

In this paper, we presented SAGE, a post-hoc method to extract interpretable, global explanations for NLP. A key ingredient of SAGE is the fact that it produces rules based on semantic, more abstract information. We focused on Named Entity Recognition, a particularly challenging NLP problem, which was under-investigated in the scope of explainability. We compared our model against different SP-LIME variants on two BERT models trained in two popular NER benchmarks: CoNLL03 and Ontonotes. Results indicate that our global explanations are more compact (hence more interpretable) and significantly better at mimicking the black-box classifier, although SAGE fails in making explanations of few particularly underrepresented classes. We also prove empirically that semantic and syntactic information bring a major boost to the explanation quality.

In the future we plan to investigate possible strategies to guarantee the production of explanations for any class, regardless the amount of predictions of such entity. In particular, we plan to combine local approaches, such as LIME, to SAGE.

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

## A  SCALABILITY

Data mining approaches are combinatorial, therefore the complexity grows non-linearly with the increase of the data. One may wonder how Fp-growth adjustments, rules pruning and selection process of the algorithm contribute in controlling the potential explosion. To shed some light on this, we report in Table 2 the number of elements produced by each step in the Ontonotes dataset (the largest corpus in our experiments).

| | |
|---|---|
| **SAGE** | 0:**synset**:Asian $\implies$ NORP
0:**synset**:European $\implies$ NORP |
| **SAGE-words** | 0:word:Chinese $\implies$ NORP
0:word:American $\implies$ NORP |
| **SP-LIME-1** | 0:word:Chinese $\implies$ NORP
0:word:American $\implies$ NORP |
| **SP-LIME-2** | 2:word:. , 0:word:Chinese $\implies$ NORP
3:word:., 0:word:American $\implies$ NORP |
| **SP-LIME-3** | 4:word:., 1:word:North, 2:word:Koreans $\implies$ NORP
-1:word:and, 16:word:., 0:word:American $\implies$ NORP |
| **SP-LIME-4** | 1:word:position, -1:word:the, 2:word:., 0:word:American $\implies$ NORP
-2:word:and, -3:word:television, 3:word:., 0:word:Palestinian $\implies$ NORP |
| **SP-LIME-5** | 8:word:., -1:word:with, 6:word:intransigence, 1:word:North, 2:word:Koreans $\implies$ NORP
-2:word:Russians, -1:word:and, -4:word:Europeans, 2:word:., 0:word:Chinese $\implies$ NORP |

Tab. 3: Two examples of generated rules per model for NORP entity in Ontonotes dataset. Only **SAGE** exploits semantics, in this case the **synset** features are important to build rules that are more generic. We can also notice how, the increase of $k$ in **SP-LIME** models makes the explanation more and more wired to the example that generated it.

| Entity | #tokens | #generated candidates | #pruning | #selection |
|---|---|---|---|---|
| EVENT | 351 | 57091 | 3078 | 90 |
| WORK_OF_ART | 594 | 304724 | 7781 | **100** |
| LAW | 222 | 23255 | 1569 | 63 |
| LANGUAGE | 22 | 0 | 0 | 0 |
| NORP | 1411 | 14216 | 1926 | 78 |
| FAC | 398 | 127311 | 5205 | 84 |
| LOC | 504 | 36894 | 2180 | 30 |
| PRODUCT | 236 | 914 | 109 | 7 |
| TIME | 668 | 10184 | 2327 | 53 |
| PERCENT | 1483 | 181897 | 26277 | 29 |
| MONEY | 1109 | 46497 | 6871 | 66 |
| QUANTITY | 463 | 18337 | 1926 | 37 |
| ORG | 5604 | 122722 | 65749 | **100** |
| GPE | 4254 | 411140 | 24738 | 53 |
| PERSON | 4151 | 43143 | 7746 | 80 |
| DATE | 4739 | 22347 | 6996 | **100** |
| CARDINAL | 1795 | 14137 | 1621 | **100** |
| ORDINAL | 322 | 488 | 91 | 4 |
| | | | | |
| ALL | 28326 | 1435297 | 166190 | 1074 |

Tab. 2: Number of elements produced by each step. In bold classes were the number of selected explanations was interrupted because the $maxnrules$ was reached.

