# OpenReview forum: "SAGE: Semantic-Aware Global Explanations for Named Entity Recognition"
_ICLR.cc/2023/Conference — Submitted to ICLR 2023_

### Official Review · Reviewer_RoTP · 2022-10-25

**Confidence:** 4
**Correctness:** 4
**Technical Novelty And Significance:** 3
**Empirical Novelty And Significance:** 3
**Recommendation:** 5

**Clarity, Quality, Novelty And Reproducibility:**

The paper is mostly clearly written and easy to follow though there are few minor things that could help such as (1) explicitly mentioning Y are the blackbox predictions and not gold truth labels ( this is apparent in Fig 1, but not in section 3.1 ), (2) the last sentence of Section 3.1 right before Algorithm 1 seems cut off (3) defining minimum support and minimum confidence explicitly, etc.  The work could be reproduced for the most part.

**Strength And Weaknesses:**

**Strengths:**
Although the use of rule sets is quite common in tabular data, its less prevalent in NLP and I don’t recall seeing the construction of rule set classifiers for explaining sequence level tasks before so its quite novel ( though the authors mention Bayesian Rule Lists which is less efficient method ).

**Weakness: **
The paper's argument could be made stronger.  The baseline they compare against seem pretty weak ( they allude to its limitations in the related work section ) and it seems like comparing SAGE with additional methods such as Decision trees would have more sense as well ( since SP-LIME performs very poorly in the task ).  More robust comparison of baselines/alternative methods  ( bayesian rules, global explain models, decision trees for NLP etc ) would benefit this paper along with some ablations including whether if instead of filtering as aggressively and using all rules if they'd get you 91% F1 on CoNLL?  If so, it seems perfectly acceptable as a solution if only a sparse amount of rules are used at each's instance explanation level.

Discussing what it means for overlapping spans ( unigram, bigram, etc ) of the same word to be present in rules is also important.  Also qualitative analysis of rules extracted including human judgement on whether they are good/interpretable, because the only rules shown in the paper are of simple unigram words/features implying a NORP class.

Finally, the accuracy of the BERT NER system on both tasks should be provided because thats really the upper limit of how well any explanation model could do.

**Summary Of The Paper:**

The authors of this work propose SAGE, a data mining method to discover global explanation rules for black box text classifiers using semantic augmented features in a post-hoc fashion.  The method first leverages FP-growth (Han et al 2000) to generate candidate implication rules of the form (t1, t2, .. tn) -> y  ( where t1, t2, etc delineate the presence of tokens and semantic features within a window and y is a class for a given task ).  They then show how to filter the generated set of rules and select a final concise set of rules.  They evaluate SAGE on the NER task, by training BERT on the task and using SAGE to explain its predictions.  They compare SAGE against SP-Lime-k which is global submodular based variant of LIME and an ablated version of SAGE which removes semantic augmentation.  They show SAGE outperforms the other methods on the CoNLL03 and OntoNotes datasets, semantic augmentation increases accuracy and additionally provide some examples of rules the system learns which highlights the utility of adding semantic augmentation.

**Summary Of The Review:**

Its an interesting idea overall, but feels under explored in the paper and could be made stronger and more compelling if it had more robust comparison of baselines/alternative methods, ablations and qualitative analysis which really highlighted why the method should be used.

---

> ### Author Response · Authors · 2022-11-19
> **Response to RoTP**
>
> All the concerns were addressed in the General Response.
> We would like to thank you again for the suggestions and feedbacks that we believe were very interesting and helped improving the quality of our paper.

---

### Official Review · Reviewer_rFXT · 2022-10-25

**Confidence:** 3
**Correctness:** 1
**Technical Novelty And Significance:** 2
**Empirical Novelty And Significance:** 1
**Recommendation:** 3

**Clarity, Quality, Novelty And Reproducibility:**

The paper is fairly easy to read. However, most terms used in the algorithms are not defined before using them, and one has to refer to subsequent sections to understand their meaning. This reduces the readability of the paper.
The approach is seems relatively novel, but it is hard to appreciate given the lack of baselines.
There is limited details about the implementation and it is not clear if one can reproduce the results in the paper.

**Strength And Weaknesses:**

The authors have provided motivation to the problem, which is certainly important. They present their algorithm and their experiments on 2 datasets, compared against the SP-Lime baseline model.

The paper could have been improved by comparison to other baselines. Given that LIME and SP-Lime were released in 2016, more work has gone into the field of XAI. It is not clear why only LIME was chosen as the baseline.


**Summary Of The Paper:**

The authors present SAGE or SEMANTIC-AWARE GLOBAL EXPLANATIONS model specifically for handling named entity recognition problems. They present a method to produce highly interpretable global rules to explain NLP classifiers.

**Summary Of The Review:**

The authors present SAGE specifically for handling named entity recognition problems and present a method to produce rules to explain NLP classifiers. The paper could have been improved by comparison to other baselines.

---

> ### Author Response · Authors · 2022-11-19
> **Response to rFXT**
>
> > The paper is fairly easy to read. However, most terms used in the algorithms are not defined before using them, and one has to refer to subsequent sections to understand their meaning. This reduces the readability of the paper.
>
> Please refer to the general response, we applied multiple changes to improve the readability of the paper.
>
> > The approach is seems relatively novel, but it is hard to appreciate given the lack of baselines.
> > The paper could have been improved by comparison to other baselines.
>
> We introduced Decision Trees as an additional baseline. Please look at the general response for further details.

---

### Official Review · Reviewer_WCcH · 2022-11-02

**Confidence:** 4
**Correctness:** 4
**Technical Novelty And Significance:** 3
**Empirical Novelty And Significance:** 3
**Recommendation:** 6

**Clarity, Quality, Novelty And Reproducibility:**

The writing is clear and the proposed approach is classic but its application is novel.
It demonstrates an effective approach to generating a global explanation for the NER model, which is unexplored.
I have a minor concern about the reproducibility.

**Strength And Weaknesses:**

Proposed approach shows its effectiveness in generating a global explanation for the NER model. I believe this approach can be useful to the future research of weakly-supervised learning and neural-symbolic learning for NER. However, I have a minor concern that the paper needs more analysis to convince readers. For example,

(1) Could find that the F1 score between model prediction and rule-based prediction is over 0.4 with only one rule. It could be good to show qualitative examples of which rules are extracted “globally”, not just for the specific entity type (e.g., NORP).

(2) Comparison of qualitative explanation examples between SAGE and LIME. Seems the word itself contributes a lot to labeling decision (e.g., Chinese → NORP), compared to the surrounding words or context. Then, I think word importance generated by LIME can also be a good explanation but it seems not. It would be good to show qualitative explanation examples between SAGE and LIME to show the effectiveness of SAGE.

(3) How does each semantic/syntactic information (POS tag, entity types, is_digit, … etc.) contribute to the performance? Seems entity prediction gives a lot of information to create rules.

(4) What is the performance of the fine-tuned BERT model as it is?


**Summary Of The Paper:**

Paper presents a post-hoc method to produce interpretable global rules to explain NER classifiers. Rules are extracted with a data mining approach that gathers labeling rule patterns by FP-growth algorithms, prunes the rules by removing soft-duplicated rules, and selects rules that maximizes the F1 score. Selected rules serve well as a post-hoc global explanation for the NER model and show its better explanation quality than LIME-based global explanation.

**Summary Of The Review:**

Paper presents a post-hoc method to produce interpretable global rules to explain NER classifiers and show its effectiveness.
Although the paper needs more analysis to convince readers, I believe this approach can be useful to the future research of weakly-supervised learning and neural-symbolic learning for NER. I'm willing to increase the score when analysis is provided.

---

> ### Author Response · Authors · 2022-11-19
> **Response to WCcH**
>
> > (3) How does each semantic/syntactic information (POS tag, entity types, is_digit, … etc.) contribute to the performance? Seems entity prediction gives a lot of information to create rules.
>
> We thank you for this comment, we believe that investigating the importance of each feature type (pos, synset, entity..) can be an interesting study, and we will deep dive on the matter. Nonetheless, SAGE being based on a data mining approach, naturally auto-selects the optimal subset of features that is statistically more relevant for the problem. Therefore, this aspect is inherently partially addressed by the algorithm itself.
>
> The rest of the comments were addressed in the general response.

---

### Official Review · Reviewer_4Q6v · 2022-11-04

**Confidence:** 2
**Correctness:** 2
**Technical Novelty And Significance:** 3
**Empirical Novelty And Significance:** 3
**Recommendation:** 3

**Clarity, Quality, Novelty And Reproducibility:**

#### Clarity:
The paper is not easy to read for me.

#### Quality:
The paper needs major revision in its delivery. Many details are skipped or just refer to figures, tables, or algorithm blocks. I’m not sure how the F1 metric is computed in the end. Does it mean that the authors only feed the black-box classifier the explanations and use the NER F1 score to evaluate explanations?

#### Novelty:
The paper misses two lines of papers, (1) papers about explanation in NLP while claiming the method is general to explanations for NLP tasks (as listed in the weaknesses), (2) papers about another line of works often also considered as local explanation, Shapley value.
Lloyd S Shapley. “A value for n-person games.”
Erik Strumbelj and Igor Kononenko. “An efficient explanation of individual classifications using game theory.” The Journal of Machine Learning Research, 11:1–18, 2010

#### Reproducibility:
The paper may not be easy to reproduce in its current status.

**Strength And Weaknesses:**

### Strengths:
* The proposed method, SAGE, is potentially useful.

### Weaknesses:
* The paper may need major revision in its writing.
  * The paper is not easy to read for me, especially the method section and its notations without complete definitions, e.g., mingensup, mingencon, maxrulelen, maxnrules, SORT-SUPPORT-AND-ABSTRACTION, etc. I need to guess their meanings.
  * The paper can be further proofreading. For example,
    * in Section 2, “Finally, Performing…” has a wrong uppercase.
    * In Section 3.1, “..explanation method, the In Algorithm 1 the …”
  * Most of the citations in this paper are not in a correct format. Please change \cite{} to \citep{}.
* The paper lacks discussion about other explanations for NLP works while this paper claims to be a general method for NLP but test on NER, such as (I'm not asking you to cite them but would like to see its position in explanation for NLP field):
  * David Alvarez-Melis and Tommi Jaakkola. “A causal framework for explaining the predictions of black-box sequence-to-sequence models.” In EMNLP 2017.
  * Hanjie Chen and Yangfeng Ji. “Learning variational word masks to improve the interpretability of neural text classifiers.” In EMNLP 2020.
  * Yi-Lin Tuan, Connor Pryor, Wenhu Chen, Lise Getoor, and William Yang Wang. "Local explanation of dialogue response generation." In NeurIPS 2021.
  * Hanjie Chen, Song Feng, Jatin Ganhotra, Hui Wan, Chulaka Gunasekara, Sachindra Joshi, and Yangfeng Ji. “Explaining neural network predictions on sentence pairs via learning word-group masks.” In NAACL 2021.


**Summary Of The Paper:**

This paper proposes SAGE, a method that extracts semantic-aware global explanations that can be applied to NLP in general and specifically experiments on name entity recognition.

**Summary Of The Review:**

My major concerns are that (1) the paper is not written well to deliver its ideas and results, and (2) according to the paper’s statement to be general in NLP, I would anticipate to see discussion about other works on explanations of NLP in this paper, which are not mentioned. I would raise my score if these are clarified.

---

> ### Author Response · Authors · 2022-11-19
> **Response to 4Q6v**
>
> >  (1) the paper is not written well to deliver its ideas and results, and (2) according to the paper’s statement to be general in NLP, I would anticipate to see discussion about other works on explanations of NLP in this paper, which are not mentioned.
>
> We addressed both your concerns. As also described in the general response, we had an important rewriting of the algorithm description and related works. In particular, in related works we better frame our approach with respect to other techniques on explanations in NLP. The works pointed out by you are relevant to the subject and helps in understanding the novelties introduced in our work. None of the mentioned papers approach the explainability in NER nor uses semantic features for producing better explanations.

---

### Official Review · Reviewer_3Fgr · 2022-11-04

**Confidence:** 4
**Correctness:** 3
**Technical Novelty And Significance:** 2
**Empirical Novelty And Significance:** Not applicable
**Recommendation:** 3

**Clarity, Quality, Novelty And Reproducibility:**

Clarity of the paper is a bit rocky.  They use some phrases repeatedly in multiple sections but do not elaborate them . Eg.They claim one caveat of LIME is that explanations must be positional. But their method is also dependent on a fixed contextual window .
Novelty exists , in the transforming of the data into a data mining task and incorporating semantic information.
Reproducibility is good since they list the hyperparameters used for the datasets mentioned in the paper. But it is unclear how much of finetuning of hyperparamers took place and how resilient the method is to any change in these values and its effectiveness for a different dataset or different task.

**Strength And Weaknesses:**

Strengths:

	1. They support the necessity for each step in the algorithm
	2. They provide global explanations
	3. They are task agnostic , method can be applied to many NLP tasks

Weaknesses:

	1. In section 3.1 they do not specify what certain notations mean  , eg the difference between the two transaction tables on the right of figure 2.
	2. Jump from section 3.2 to 3.3 is big especially for people who are unfamiliar with algorithms they point to such as FP-growth Han et al. (2000) and apriori Agrawal et al. (1994). They use an example for section 3.1 but then they drop the example for subsequent sections in the algortihm .
	3. Other evaluation metrics employed by other papers eg, fidelity to the model and comprehensibility could have been explored . Human evaluations might make a more compelling case .
	4. They don’t perform any study about which semantic features help and which harm the f1 score.
	5. Visualizaition is an important part of explainable models which this paper lacks



**Summary Of The Paper:**

In this paper the authors introduce a new post-hoc method to create explanations for a black box classifier. They explain the steps in the method using the task of Named Entity Recognition as example. They turn the data into baskets of information , by extracting the contextual window around each word as well as some semantic features like pos tags, synsets etc and then use a data mining algorithm to extract patterns of co-occurrence corresponding to each classification label. Pruning is done over these patterns or candidate explanations based on thresholds and explanations are then selected greedily using f1 score maximization. The perform evaluation on this method by measuring the F1 score vs complexity defined by the number of explanations and they compare this to variations of LIME as well as a version of their model without semantic information


**Summary Of The Review:**

Although the fundamental idea is good, it fails to convince that the method is robust across different datasets or NLP tasks.
They employ a contextual window of +/-3 tokens and that might not be transferrable to other NLP classification tasks like sentiment classifier or recommendation systems.

---

> ### Author Response · Authors · 2022-11-19
> **Response to reviewer 3Fgr**
>
> > 1. In section 3.1 they do not specify what certain notations mean  , eg the difference between the two transaction tables on the right of figure 2.
> >2. Jump from section 3.2 to 3.3 is big especially for people who are unfamiliar with algorithms they point to such as FP-growth Han et al. (2000) and apriori Agrawal et al. (1994). They use an example for section 3.1 but then they drop the example for subsequent sections in the algortihm .
>
> We understand the algorithm description can be difficult for people unfamiliar with apriori-like algorithms. We tried to mitigate it by rearranging the Section and by adding some details. We also introduced missing notations. Concerning figure 2, the caption was improved, it should be clearer now.
>
> > 3. Other evaluation metrics employed by other papers eg, fidelity to the model and comprehensibility could have been explored . Human evaluations might make a more compelling case .
>
> As we specified also in the general comment, the F1-score terminology was misleading, we were in fact measuring fidelity. Again we apologize for that and changed the terms accordingly.
> As a measure of comprehensibility instead, we used complexity (#explanations per class), since our rules are composed of few condition terms. This principle, already used in literature, directly correlates with the comprehensibility of explanations.
> Human evaluations would always be the best method to measure the explanations quality, however they are difficult to collect.
>
> > 4. They don’t perform any study about which semantic features help and which harm the f1 score.
>
> We thank you for this comment, we believe that investigating the importance of each feature type (pos, synset, entity...) can be an interesting study, and we will deep dive on the matter. Nonetheless, SAGE being based on a data mining approach, naturally auto-selects the optimal subset of features that is statistically more relevant for the problem. Therefore, this aspect is inherently partially addressed by the algorithm itself.
>
>
> > 5. Visualizaition is an important part of explainable models which this paper lacks
>
> Please see the general response comment.
>
>
> > They use some phrases repeatedly in multiple sections but do not elaborate them . Eg.They claim one caveat of LIME is that explanations must be positional. But their method is also dependent on a fixed contextual window .
>
> I believe you are referring to those statements in the paper:
> > Furthermore, token classification tasks, such
> as NER, introduce additional complexity because explanation must be positional…
>
> >LIME is widely used for explaining models in classification problems. However,
> there are some caveats when applying it in NER. Indeed, predictions are at token-level, while usually the approach is applied for document-level classification problems. Thus, explanations must be positional.
>
> Positionality of the explanations is required by the NER task which is a token level classification problem, it is not a LIME constraint. Positionality is paramount in NER, because the token-level decision usually depends on itself and its local context.
> The problem is that adapting LIME to provide token-level, positional explanations introduces some caveats.
>
>
>
> > Although the fundamental idea is good, it fails to convince that the method is robust across different datasets or NLP tasks. They employ a contextual window of +/-3 tokens and that might not be transferrable to other NLP classification tasks like sentiment classifier or recommendation systems.
>
> Our approach is general and holds to any text classification problem, such as sentiment analysis or recommended systems. We have put the accent on NER because it is underexplored in the frame of explainability.
> The contextual window, along with the necessity of having positional features are specific to Named Entity Recognition, or more in general to any token classification task. In traditional text classification problems they are not needed, hence the feature construction process is actually simplified.
> In practice, in text classification we only need to use a different feature function $\Phi$ that maps text in a bag-of-word representation (which is straightforward), then the approach is analogous.

---

### Author Response · Authors · 2022-11-18
**Summary of the Updates and General Response**

We would like to thank the reviewers for their comments. We really appreciated the recognition of technical and empirical novelty of our work (**WCcH**, **RoTP**, **4Q6v**). We also would like to thank the authors for their concerns and feedback, they were relevant and we agreed on the importance of certain aspects that we improved in the revision. We believe the revised version addresses all of those issues, which greatly benefitted the overall paper quality.


### Summary of the updates and responses
Here the summary of the updates made in the draft:

**Writing Quality**   ( **4Q6v**, **3Fgr**)
We revised related works and contextualized our model also with respect to other NLP papers as pointed out by **4Q6v**.
We improved the description of the SAGE algorithm, and defined all the used symbols.
We corrected minor issues indicated by the reviewers.
Clarification of the evaluation metrics were applied in the paper. See **Evaluation Criteria**.

**Visualization**  (**3Fgr**,**WCcH**, **RoTP**)
We understand visualization is a crucial aspect of explainable models, therefore we included a full example of generated rules instead of focussing on single entity class (NORP). Table 1 in the old version was moved in Appendix and it was replaced by Figure 5 in the revised version.
Sketch of the algorithm (Figure 1) was revised too, making it more intuitive.

**Evaluation Criteria**  (**3Fgr**,  **WCcH**, **RoTP**, **4Q6v**)
We realized that the usage of F1 score terminology was misleading and created some confusion. What we are in fact measuring is the fidelity of the explanations with respect to the black-box classifier. Hence we changed the term F1 with fidelity, which is more appropriate and intuitive.
This is why we do not report the performance of the fine-tuned BERT model, since they are not relevant nor of interest to our experimentation.

**Baselines comparison** (**RoTP**, **rFXT**)
 Some reviewers pointed out the lack of strong baselines to compare our model with. Although LIME is not the most recent local explanation technique, we chose it because it is well established and widely used for text classification, and it also provides a method to obtain global explanations.
Nonetheless, we agree on the fact that adding other baselines is a fundamental aspect.

Following the suggestion of **RoTP**, we introduced another baseline: a Decision Tree (DT) classifier.
We believe the choice of DT completes the picture because it allows us to compare our model against two orthogonal approaches: LIME (a __local explanation__ method) and DTs that are explainable by design, __global explanation__ algorithms.
Thanks to the introduction of DTs,  we were also able to observe the benefits of semantics on a different global explanation algorithm. Although this was not surprising, it further highlighted one of the contributions of our paper.

**Note**: We will address author-specific and minor concerns in dedicated comments.

---

### Decision · Program_Chairs · 2023-01-20

**Decision:**

Reject

**Justification For Why Not Higher Score:**


There baselines are weak, and the paper needs additional analysis work.

**Justification For Why Not Lower Score:**

n/a

**Metareview: Summary, Strengths And Weaknesses:**

The paper studies explainability in NER.
It puts forward a   post-hoc method to generate interpretable global rules to explain NER classification decisions.
The reviewers agree that approach can  potentially be impactful.  However, in its current form it requires stronger baselines from the literature, ablations and qualitative analysis to help convince the reader why the approach is useful.  The authors are encouraged to continue their work, address reviewer concerns, and  resubmit at a future conference at ICLR or other venues.

**Summary Of Ac-Reviewer Meeting:**

n/a